# From the Bush to the Brain: Preclinical Stages of Ethnobotanical Anti-Inflammatory and Neuroprotective Drug Discovery—An Australian Example

**DOI:** 10.3390/ijms241311086

**Published:** 2023-07-04

**Authors:** Payaal Kumar, Shintu Mathew, Rashmi Gamage, Frances Bodkin, Kerrie Doyle, Ilaria Rossetti, Ingrid Wagnon, Xian Zhou, Ritesh Raju, Erika Gyengesi, Gerald Münch

**Affiliations:** 1Pharmacology Unit, School of Medicine, Western Sydney University, Campbelltown, NSW 2560, Australiaauntie.fran80@gmail.com (F.B.); i.rossetti@westernsydney.edu.au (I.R.); i.wagnon@westernsydney.edu.au (I.W.);; 2Indigenous Health Unit, School of Medicine, Western Sydney University, Campbelltown, NSW 2560, Australia; auntykerrie.doyle@westernsydney.edu.au; 3NICM Health Research Institute, Western Sydney University, Westmead, NSW 2145, Australia

**Keywords:** neuroinflammation, Alzheimer’s disease, anti-inflammatory drugs, metabolites, drug discovery, pre-clinical drug discovery

## Abstract

The Australian rainforest is a rich source of medicinal plants that have evolved in the face of dramatic environmental challenges over a million years due to its prolonged geographical isolation from other continents. The rainforest consists of an inherent richness of plant secondary metabolites that are the most intense in the rainforest. The search for more potent and more bioavailable compounds from other plant sources is ongoing, and our short review will outline the pathways from the discovery of bioactive plants to the structural identification of active compounds, testing for potency, and then neuroprotection in a triculture system, and finally, the validation in an appropriate neuro-inflammatory mouse model, using some examples from our current research. We will focus on neuroinflammation as a potential treatment target for neurodegenerative diseases including multiple sclerosis (MS), Parkinson’s (PD), and Alzheimer’s disease (AD) for these plant-derived, anti-inflammatory molecules and highlight cytokine suppressive anti-inflammatory drugs (CSAIDs) as a better alternative to conventional nonsteroidal anti-inflammatory drugs (NSAIDs) to treat neuroinflammatory disorders.

## 1. Chronic Neuroinflammation and Its Role in Alzheimer’s Disease

Neuroinflammation refers to inflammation that occurs in the brain in response to injury, infection, disease, or other harmful stimuli. Inflammation is a normal response of the immune system to protect against harmful stimuli, such as pathogens, toxins, or tissue damage. However, when the inflammatory response becomes chronic, it can contribute to various neurological disorders, including Alzheimer’s disease (AD), Parkinson’s disease (PD), Multiple sclerosis (MS), and—as a reaction to injury—to the damage after a stroke [1,2,3,4]. Neuroinflammation involves the activation of various immune cells in the brain, including microglia, astrocytes, and peripheral immune cells, such as monocytes, that can cross the blood–brain barrier. These cells release inflammatory mediators, such as cytokines, chemokines, and reactive oxygen species (ROS), that can damage brain structures and alter neurotransmitter signaling, leading to cognitive and behavioral changes [5].

Multiple sclerosis has long been recognized as the ‘classical’ neuroinflammatory disease [6]. MS is a chronic autoimmune disease that affects the central nervous system. T cells play a key role in the development of MS. In MS, T cells attack and damage the myelin sheath that surrounds and protects nerve fibers in the brain and spinal cord. This leads to inflammation and scarring (sclerosis) of the affected areas, which can disrupt or block nerve signals, causing a wide range of symptoms. T cells are involved in both the initiation and progression of the disease, and their activation and proliferation are thought to be important in driving the autoimmune response in MS [7].

More recently, the term ‘neuroinflammation’ has been applied to chronic, Central nervous system (CNS)-specific, inflammation-like glial responses that do not reproduce the classic characteristics of inflammation and predominantly involve the innate immune system of the brain, including microglial and astroglia [8]. Chronic microglial activation has been described in many neurodegenerative diseases, such as chronic traumatic encephalopathy (CTE), amyotrophic lateral sclerosis (ALS), Alzheimer’s disease (AD), and Parkinson’s disease (PD). It is believed that neuroinflammation plays a key role in the development and progression of these diseases, and targeting neuroinflammation has become a promising area of research for the development of new treatments [9,10].

In the last decades, it has become obvious that the innate immune system plays a key role in the development and progression of AD [11,12]. In AD, the innate immune system is activated in response to the accumulation of beta-amyloid plaques and tau-containing tangles as well as debris from dying and degenerating neurons in the brain. These abnormal protein aggregates add intracellular components off neurons (damage-associated molecular pattern molecules, DAMPs) and can trigger the activation of microglia, the resident immune cells of the central nervous system (CNS). Once activated, microglia release a variety of pro-inflammatory molecules, such as cytokines and chemokines, which can contribute to the inflammation and nerve cell death seen in AD [13].

Debris released from dying neurons includes DAMPs. DAMPs are released from damaged neurons to alert the innate immune system and activate several signal transduction pathways through interactions with the highly conserved pattern recognition receptors (PRRs). DAMPs activated pro-inflammatory pathways, mediating the release of free radicals and pro-inflammatory cytokines. DAMPs include amyloid β (Aβ), high-mobility group box 1 (HMGB1), the S100 family proteins, chromogranin A, and nucleic acids [14].

Furthermore, research has identified potent DAMP-like microglial activators, called advanced glycation endproducts (AGEs), and their receptor (RAGE) as an additional important inflammatory pathway [15]. AGEs bind to the RAGE receptor and cause several inflammatory processes to be triggered. Although AGE accumulation in cells and tissues is a typical aspect of aging [16], it is increased in AD [17,18]. Pathological AD deposits, such as amyloid plaques and neurofibrillary tangles, can be found to contain AGEs [19,20]. AGE modification of proteins explains many of the neuropathological and biochemical features of AD, such as extensive protein crosslinking, glial induction of neuroinflammation and oxidative stress, and excess neuronal cell death [21,22].

In the presence of the abovementioned pro-inflammatory stimuli, microglia transform to an activated phenotype releasing inflammatory factors, such as pro-inflammatory cytokines and free radicals. For example, microglial activation, accompanied by increased levels of pro-inflammatory mediators, such as Tumor necrosis factor-α (TNF-α), Interleukin-1β (IL-1β), and IL-6, prostaglandins, and reactive oxygen and nitrogen species, is observed in the AD brain at all stages of the disease [23]. The activation of microglia also stimulates the production of reactive oxygen species (ROS) and reactive nitrogen species (RNS), including nitric oxide (NO), which can lead to oxidative stress and cause further damage to cells in the brain [24].

IL-6 is one of the most intriguing cytokines found in AD brains. Cortical senile plaques in AD patients exhibit significant IL-6 content, but no Il6 positivity was discovered in normal brains, according to Huell et al. [25]. In addition, dementia can be also correlated to high levels of systemic IL6. For example, elevated serum IL-6 levels in midlife predict cognitive decline; the combined cross-sectional and longitudinal effects over the 10-year observation period corresponded to an age effect of 3.9 years [26]. Furthermore, data collected over 20 years from 2422 participants in the “Epidemiology of Hearing Loss Study”, demonstrated that patients with cognitive impairment had higher IL-6 levels on average [27].

Furthermore, genome-wide association studies (GWAS) have identified three inflammation-relevant genes that are linked to AD, including clusterin (CLU), complement receptor 1 (CR1), and triggering receptor expressed on myeloid cells 2 (TREM2) [28].

In addition to microglia, astroglia may also play a role in the pathology of AD through the process of neuroinflammation and the release of cytokines. Research has shown that astroglia can become activated in response to the accumulation of beta-amyloid plaques, a hallmark of AD, and release inflammatory cytokines that can contribute to neuronal damage and cognitive decline [29]. Astroglia is also transformed to a reactive state, thereby neglecting their neuro-supportive functions, thus rendering neurons vulnerable to neurotoxins, including pro-inflammatory cytokines and reactive oxygen species (“neuro-neglect hypothesis”) [30].

In summary, all these findings suggest that the progression of many neurodegenerative diseases, including AD [31], is—at least partly—driven by a cycle of self-perpetuating inflammatory neurotoxicity according to the following steps [32].

(a)Various inflammatory triggers can cause the initial activation of microglia. These triggers can be peripheral (e.g., systemic infections or peripheral chronic inflammation) or central (e.g., trauma, degenerating/dying neurons, or amyloid deposits) [33]. In our GFAP-IL6 mouse model, the trigger is the brain-specific production of the cytokine IL6 [34,35].(b)Activated microglia release neurotoxic factors, including cytotoxic cytokines (such as TNF-α) and reactive oxygen and nitrogen species, which cause damage to neighboring neurons.(c)These damaged neurons release various microglia activators, such as damage-associated molecular pattern molecules (DAMPs) [36], which results in further microglial activation. Consequently, targeting chronic neuroinflammation has been suggested as a disease-modifying treatment for many neurodegenerative diseases, including AD.

## 2. Cytokine Suppressive Anti-Inflammatory Drugs—A Better Alternative to Conventional Anti-Inflammatory Drugs?

### 2.1. Nonsteroidal Anti-Inflammatory Drugs (NSAIDs)

Nonsteroidal anti-inflammatory drugs (NSAIDs) are the mainstay of treatment for inflammation and pain. NSAIDs are used to treat conditions, such as osteoarthritis, rheumatoid arthritis, and other chronic degenerative inflammatory conditions, as well as to reduce pain, fever, and inflammation associated with different types of injuries or surgeries. NSAIDs work by inhibiting the activity of cyclooxygenases (COX), which are responsible for the production of prostaglandins. Prostaglandins are hormone-like molecules that play a critical role in the inflammatory response. There are two types of COX enzymes, COX-1 and COX-2. COX-1 is responsible for the production of prostaglandins that are involved in maintaining the normal function of the stomach, kidneys, and platelets. COX-2 is responsible for the production of certain prostaglandins that are involved in the inflammatory response. There are different types of NSAIDs, such as traditional nonselective (ibuprofen, naproxen) and selective COX-2 inhibitors (celecoxib, etoricoxib) [37].

There is some evidence to suggest that NSAIDs may have a protective effect against AD. Several observational studies have found that regular use of NSAIDs is associated with a lower risk of developing AD [38,39]. These studies suggest that the anti-inflammatory properties of NSAIDs may play a role in protecting against the development of AD. Additionally, clinical trials have been conducted to test the effects of NSAIDs on AD, but the results have been inconclusive. Some studies have found that NSAIDs can slow the progression of AD, while others have found no effect or even negative effects [40]. Given the inconsistent results of these studies, it is not currently recommended to use NSAIDs as a preventive measure for AD. It is important to note that despite the efficacy of NSAIDs in treating inflammation in general, long-term use of these drugs also can cause gastric and renal side effects [41].

However, one of the reasons why NSAIDs do not decrease neuroinflammation and are ineffective in neuroinflammatory disorders is that they do not interfere with the production of chemokines and cytokines at therapeutic concentrations. NSAIDs are designed as specific inhibitors of cyclooxygenases (COXs) and targeted to only decrease the production of prostaglandins, and not of other pro-inflammatory mediators. This limitation on one particular inflammatory target might be the reason for their poor efficacy in treating neurodegenerative diseases with a neuroinflammatory component.

### 2.2. Steroidal Anti-Inflammatory Drugs (Corticosteroids)

Corticosteroids are a class of hormones produced by the adrenal cortex. These hormones play a vital role in regulating the body’s immune response, metabolism, and stress response. Synthetic versions of these hormones, such as prednisone, dexamethasone, and hydrocortisone, are used as medications to treat a wide range of conditions, including autoimmune disorders, allergies, asthma, and inflammatory diseases.

Some observational studies have found that regular use of steroidal anti-inflammatory drugs (SAIDs) is associated with a lower risk of developing AD [42,43]. In a co-twin study by Breitner et al., the onset of AD was inversely associated with prior use of corticosteroids, such as prednisone, prednisolone, methylprednisolone, dexamethasone, hydrocortisone, betamethasone, and triamcinolone (odds ratio [OR], 0.25; 95% confidence interval [CI], 0.06 to 0.95; *p* = 0.04) [42]. In the study by Nerius et al., which was based on German insurance data, the lowest risk was found among users of inhaled glucocorticoids, such as beclometason, budesonid, ciclesonid, flunisolid, fluticason, and mometason (HR = 0.65, CI = 0.57–0.75), followed by nasal (HR = 0.76, CI = 0.66–0.87), other (HR = 0.84, CI = 0.80–0.88), and oral, such a prednisolon, prednison, methylprednisolon, hydrocortisone, and dexamethason (HR = 0.83, CI = 0.78–0.88) [43]. 

In addition, post-mortem brain investigations revealed that subjects having received corticosteroids for other medical conditions had significantly lower ratings and counts of AD hallmarks, such as neuritic plaques (NPs) and neurofibrillary tangles (NFTs) in the cerebral cortex and amygdala [44]. These studies suggest that the anti-inflammatory properties of SAIDs may play a role in protecting against the development of AD. However, a randomized controlled trial of prednisone in AD with 138 subjects showed, for one year, no difference in cognitive decline between the prednisone and placebo treatment groups [45].

It is also important to note that steroids are powerful drugs with many side effects, such as the increased risk of infections, weight gain, mood changes, fluid retention, high blood pressure, osteoporosis, skin changes, glaucoma and cataracts, and increased blood sugar levels, particularly when used in high doses or for a long period [46].

In the absence of conclusive evidence for the prevention and treatment of AD with NSAIDs or steroids and the presence of severe side effects for these drugs, the search for novel anti-inflammatory drugs has steered to compounds that inhibit the release of pro-inflammatory cytokines, including those from medicinal plants and neutraceuticals [47]. This led to the discovery of cytokine suppressive anti-inflammatory drugs (CSAIDs), which have a broader range of actions; they decrease the production of pro-inflammatory cytokines such as IL1, IL6, TNF-α, or nitric oxide produced by inducible nitric oxide synthase (iNOS) [48].

### 2.3. Cytokine Suppressive Anti-Inflammatory Drugs

Cytokine suppressive anti-inflammatory drugs (CSAIDs) are a broad class of drugs that work by inhibiting the production and/or activity of pro-inflammatory cytokines. CSAIDs target these pro-inflammatory cytokines and inhibit their production or activity, thereby reducing inflammation and the associated symptoms and complications. Examples of cytokine-suppressive anti-inflammatory drugs include TNF-α and IL-1 inhibitors [49]. TNF-α inhibitors, such as adalimumab, etanercept, and infliximab, are a class of drugs that bind to TNF-α and prevent it from binding to its receptors on the surface of cells, thereby inhibiting its activity and reducing inflammation [50]. IL-1 inhibitors, such as anakinra and rilonacept, are another class of drugs that target the pro-inflammatory cytokine IL-1 [51].

In addition to direct cytokine receptor antagonists, certain CSAIDs interfere with signaling pathways linked to the expression of pro-inflammatory cytokines. Neutraceuticals, such as curcumin, apigenin, docosahexaenoic acid, epigallocatechin gallate, α-lipoic acid, and resveratrol, have been identified to possess anti-inflammation and are considered prototype CSAIDs [52,53,54]. Both curcumin and apigenin exert a broad range of anti-inflammatory effects; they also penetrate the BBB in animal models and are safe (curcumin: generally regarded as safe (GRAS) by the FDA, apigenin: GRAS as the major ingredients of parsley and chamomile extract). Apigenin (4′,5,7-trihydroxyflavone) is a flavonoid found in chamomile, celery, grapefruit, and parsley (up to 0.5% wet weight in parsley). Apigenin has been isolated from the Australian native mint *Mentha australis* R.Br., which has been traditionally utilized by the aboriginal people for herbal medicine remedies [55]. Apigenin and its derivatives have also been isolated from *Tasmannia lanceolata*, an aromatic shrub originating from Tasmania. *T. lanceolata* leaves, berries, and barks were utilized as herbs and therapeutic agents by indigenous communities [56]. Apigenin inhibits IL-6, TNF-α, and nitric oxide production in microglia at low micromolar concentrations (IC_50_ = approx. 4 µM) [57]. Apigenin enters the brain, reaching a concentration of 1.2 µM after daily i.p. administration of 20 mg/kg apigenin for one week [58]. Furthermore, a variety of studies indicate CNS effects of apigenin when delivered i.p. or orally [59]. For example, apigenin (40 mg/kg) improved memory deficits in an amyloid-based transgenic mouse model of AD, the APP/PS1 mouse [59]. Apigenin also reduces the numbers of Iba1^+^ microglia by about 40–50% both in the cerebellum and hippocampus in GFAP-IL6 mice [35]. In humans, apigenin taken orally is systemically absorbed and recirculated by enterohepatic and local intestinal pathways. The bioavailability of apigenin from dietary sources can range from less than 1% to up to 60%, with an average of around 10–20% [60]. Furthermore, apigenin exerts potent neuroprotection in the rotenone model of PD and therefore may act as an effective agent for treatment of PD [61].

The main curcuminoid in turmeric (*Curcuma longa*) is curcumin (1E,6E)-1,7-bis (4-hydroxy-3-methoxyphenyl)-1,6-heptane-3,5-dione. By inhibiting the expression of cyclooxygenase-2 (COX-2), inducible nitric oxide synthase (iNOS), TNF-, IL-1, -2, -6, -8, and -12, among others, curcumin has a wide cytokine-suppressive anti-inflammatory effect. It interferes with the initial signaling stages downstream of the IL-6 receptor in the activation of microglia by inhibiting IL-6-induced signal transducer and activator of the transcription (STAT3) phosphorylation and subsequent STAT3 nuclear translocation [9,62] and interferes with the first signaling steps downstream of the IL-6 receptor in microglial activation. Highly bioavailable curcumin formulations (encapsulated in liposomes or micelles), such as “Longvida” (VS Corp) or Meriva (Indena), can achieve micromolar concentrations in the animal brain [63,64]. In humans, oral curcumin is degraded at the alkaline pH in the intestine; it is also extensively metabolized by the gut microflora and the liver, leading to a substantial loss of free curcumin in the adsorption and metabolism steps. Many studies suggest that even the best curcumin preparation does not lead to a plasma concentration of more than 10 ng/mL of free (not conjugated) curcumin in plasma. Curcumin has gained significant attention due to its strong anti-inflammatory properties [65]. However, despite its potency, curcumin’s limited bioavailability and stability have prompted the exploration of alternative compounds, including those derived from Australian plants. Curcumin is the gold standard in the field of anti-inflammatory drug development and exhibits multifaceted molecular mechanisms that target key pathways involved in the inflammatory process, including inhibition of various pro-inflammatory enzymes and signaling molecules. However, despite these remarkable therapeutic properties, curcumin faces challenges related to its limited bioavailability and stability, hindering its optimal clinical translation. Therefore, the search for more potent and more bioavailable compounds from other plant sources is ongoing, and our short review will outline the pathways from the discovery of bioactive plants through to the structural identification of active compounds, testing for potency, and then neuroprotection in a triculture system, and finally, the validation in appropriate neuro-inflammatory mouse models (Figure 1). Overall, this pathway involves a multidisciplinary approach that combines traditional knowledge, modern technology, and various biological assays to identify and validate bioactive compounds with neuroprotective effects.

## 3. Australian Native Plants as a Source of Novel Anti-Inflammatory Drugs

### 3.1. Aboriginal Knowledge as a Source of Information about the Medical Use of Australian Plants

From time immemorial, humanity has relied on plants and animals to keep them healthy [66]. As some of the world’s oldest living collections of independent and diverse cultures, Aboriginal and Torres Strait Islander peoples on the Australian continent and islands have developed a profound connection with their country, and the native flora and fauna for more than 60,000 years, despite the challenges over the centuries by the recently formed and imposed British nation-state.

The transmission of knowledge within and across different Aboriginal communities was primarily carried out orally, with stories, songs, and ceremonies used to pass on knowledge from generation to generation. Aboriginal healers, known as “Traditional Healers”, e.g., called “Ngangkari” by the Ngaanyatjarra, Pitjantjatjara, and Yankunytjatjara (NPY) people in the remote western desert of Central Australia, possess a deep understanding of the medicinal properties of plants and other natural resources and were responsible for the care of their communities. The knowledge the Ngangkari possessed was often held secret and passed down through an apprenticeship system [67]. Regular use of medical plants in their diet and for the treatment of ailments may have helped them to live in harmony with their country and thrive in the sometimes adverse climate of Australia over the past 60,000 years [68,69]. They have been collecting pharmacological knowledge about plants for the treatment of sores, wounds, ailments, snake bites, and skin infections, which are especially common in Aboriginal communities [70]. Most of these native plants were reported to possess unique biochemical, chemical, and nutritious organoleptic characteristics due to their unique adaptation to Australia’s extreme vegetative and climatic conditions [71,72,73].

This traditional knowledge is an extremely valuable source for the identification of novel drug leads for treating various infectious and noninfectious diseases, including inflammation-related illnesses [74,75]. For example, tea tree oil, an essential oil derived from the Australian native plant *Melaleuca alternifoliavia* is traditionally used by the Bundjalung peoples who, in being connected to part of the mid-east coast of Australia, make tea for throat ailments. It has been a popular ingredient in a variety of household and cosmetic products due to its antiseptic, anti-inflammatory, broad-spectrum antimicrobial, antiprotozoal activities, and antioxidant properties [76,77]. Today, tea tree oil is marketed as a remedy for various ailments over the counter in Australia, Europe, and North America. Another example is the development of topical preparation to treat skin inflammation from the Australian Northern Kaanju (Kuuku I’yu) medicinal plant *Dodonaea polyandra* (hop bush) [78,79].

It may be argued that one of the most important works in the area of ethnobotanical drug discovery is the Dharawal Pharmacopeia (compiled by Dharawal Aunty, Dr. Frances Bodkin), an invaluable collection of medicinal knowledge derived from the Dharawal peoples of Australia (whose ancestors are directly connected to part of the southeast coast of Australia) [80]. The Dharawal peoples have long possessed an intimate understanding of their surrounding environment, observing the intricate relationships between their country, plants, animals, cultural lore/law, and human health. Their traditional healing practices, rooted in a deep respect for nature, have provided them with a wealth of knowledge on the therapeutic properties of various flora and fauna. For example, the genus *Eucalyptus* (Myrtaceae), predominantly native to Australia, has a rich history of spiritual, cultural, medicinal, and practical living uses for the Dharawal peoples, and that the *Eucalyptus* is also deeply imbedded within the storylines and song lines of many other Aboriginal nation groups throughout Australia. Traditionally, for the Dharawal peoples, the *Eucalyptus* leaf and bark extracts brewed in water as well as smoke generated from young leaves were used to relieve cold and flu symptoms. The leaf, bark, and stem extracts have been employed to ease aches and internal pain. Aqueous inner stem extracts of *Eucalyptus* have been utilized as herbal mouthwash and cleansers for alleviating mouth sores and toothache [80]. The interest of the *Eucalyptus* species in the field of pharmaceuticals, cosmetics, and agriculture has been steadily increasing in the last few decades [81]. Eucalyptol or cineole, one of the major terpenoids originating from *Eucalyptus,* has been studied at preclinical and clinical levels. The Dharawal Pharmacopeia has been instrumental to identify potent anti-inflammatory compounds from *Eucalyptus benthamii* [74], *Eucalyptus viminalis* [82], and *Angophora costata* [83]. Other reports also suggest that the pharmacological actions of *Eucalyptus* phytochemicals demonstrate potential in the treatment of respiratory disorders, pain, and cancer [81,84,85]. Furthermore, plants, such as spreading sneezeweed (*Centipeda minima*), goat’s foot (*Ipomoea pes-caprae*), snake vine (*Tinospora smilacina*), sandpaper fig (*Ficus opposite*), stinking passionflower (*Passiflora foetida*), emu bush (*Eremophila sp.*), and hop bush (*Dodonaea viscosa* and *D. polyandra*), are a few popular Aboriginal medicinal plants that contributed in the biodiscovery of promising drug candidates for anti-inflammatory treatments [69]. Chemical investigations performed on the Australian plant *Eremophila* (Myoporaceae) is another good example of the isolation of bioactive flavonoids, phenolic compounds, and many unique and structurally diverse terpenoids, particularly sesquiterpenoids and diterpenoids with many therapeutic benefits, including antibacterial, antioxidant, and anti-inflammatory properties [42,86].

It is evident from these studies that it is worthwhile to ethically revitalize and preserve the often untapped source of Aboriginal and Torres Straits peoples’ medicinal knowledge for the discovery of novel medicinal compounds in a culturally appropriate manner.

### 3.2. Ecology of the Australian Rainforest as a Further Source of Information about the Potential Medical Use of Australian Plants

Understanding the rich diversity of the tropical rainforest and its complex web of chemical interactions between plants, animals, and microbes is the basis of another search strategy for bioactive compounds. Understanding the principles of rainforest ecology helps to decode this chemical language and guide the search for new phytotherapeutic chemicals. The native Australian rainforest is unique in the world as it has been isolated from other rainforests around the world for millions of years. As a result, it has developed a unique collection of flora and fauna that cannot be found anywhere else. It consists of an inherent richness of species that have evolved in the face of dramatic environmental challenges over more than 100 million years due to its prolonged geographical isolation from other countries [87]. The evolutionary race between herbivores and plants has created a huge diversity of plant secondary metabolites and these biotic interactions are the most intense in the rainforest. It is estimated that the rainforest is home to over 3000 species of plants, many of which are endemic to the region. Natural product discovery has been reliant on the biodiversity of the planet, and the chemical diversity of rainforest flora is unparalleled. Rainforest plants in particular survive under high biotic pressures that lead to higher levels of chemical defense and the production of a greater diversity of secondary metabolites [88]. Survival in such environments is highly competitive with plants enduring high temperatures, rainfall, and humidity as well as herbivore and microbial attacks and high UV radiation [87,89]. Over time, these plants develop unique survival techniques and phytochemicals specific to their habitat and may acquire a rich source of diverse secondary metabolites [88,90,91]. Additionally, Australian rainforests appear to have an exceptionally high incidence of endemic plants, with many being found only at restricted locations, such as the extremely isolated genera *Austrobaileya* and *Idiospermum* [87].

The therapeutic potential of rainforest plants can be appreciated from notable past achievements of drug innovations originating from the rainforest. They include quinine, a bitter-tasting alkaloid derived from the alkali extract of *Cinchona* tree bark from Andean forests [92]. Quinine played a vital role in drug discovery for over two centuries and became the first widely used drug in the treatment of malaria [87,92]. Tubocurarine, another alkaloid isolated from *Chondrdendron tomentosum* from the South American rainforest, was the first typical nondepolarizing muscle relaxant [93], while cocaine, another alkaloid originating from shrub *Erythroxylum coca*, became the first effective local anesthetic [94]. A shining example of a drug from the Australian rainforest is EBC-46, a compound found in the seeds of the blushwood tree (*Hylandia dockrillii*), which is native to the tropical rainforests of Australia. Studies have shown that EBC-46 can destroy cancer cells by triggering apoptosis. The drug is administered directly into the tumor site, where it appears to target the blood vessels that supply the tumor, leading to a rapid collapse of the tumor and the surrounding tissue [95,96].

In summary, Australian rainforests have evolved under unique environmental conditions and thus they typify an outstanding diversity of flora. As a result, it is expected that these rainforests would produce higher concentrations and varieties of pharmacologically active plants and thus serve as an untapped source for novel bioactive compounds [97,98,99,100,101].

## 4. Cell-Based Screening Assays for Novel Anti-Inflammatory Drugs

Biological screening is a critical component of the drug discovery process. Molecules derived from natural products offer a great potential to be identified as novel bioactive compounds; however, the search for active compounds from crude extracts is both time-consuming and capital-intensive. For this reason, it is critical to incorporate quick but efficient high-throughput anti-inflammatory screening assays integrated within the early processes of drug discovery [102]. Here, we will discuss a couple of widely used cell models and measurable pathways that are commonly used for the screening processes. We have used these assays in a variety of studies leading to the discovery of compounds with potent anti-inflammatory activity (Table 1).

### 4.1. Macrophages as Screening Tools for Anti-Inflammatory Drugs

Macrophages are mononuclear immune cells that play a significant role in immunity and immune responses [112]. Relatively long-lived and terminally differentiated hematopoietic cells, macrophages originate from progenitors in the embryo and are seeded throughout the mammalian body. They display a wide range of membrane receptors that recognizes and responds to a large group of host-derived and foreign ligands [113]. Macrophages regulate lymphocyte activation and proliferation and are essential players in the activation process of T and B lymphocytes by antigens and allogenic cells [112].

As part of the immune response, macrophages are drawn to foreign substances due to the presence of antibodies and conduct the fundamental protective role of destroying the invader via the process of phagocytosis [114]. They can internalize their entire surface membrane in 20 min as well as recycle components concurrently to maintain membrane homeostasis [113]. In addition, within the cytokine network, macrophages are a major source of cytokine involved in immune response, inflammation, and other homeostatic processes. Upon stimulation by microbial products and other endogenous factors, macrophages can de novo synthesize and release a plethora of cytokines, such as interleukins (IL-1 and IL-6), tumor necrosis factor (TNF-α), and interferons (IFN) [115].

One of the fascinating aspects of macrophages is that these cells can be activated to use them as screening assays for anti-inflammatory drugs [114]. Depending on the type of cytokine and other mediators that macrophages are exposed to in their local microenvironment, they are subjected to either classical (Th1) or alternative (Th2) activation. Classically activated macrophages are essentially formed in response to two signals. Firstly, IFN-γ, which acts as a primer, activates transcription factors, STAT 1 and STAT 2, which bind to the GAS regulatory sequence in immune effector genes. The second activator is typically a toll-like receptor (TLR) ligand, expressed on microbial organisms, such as lipopolysaccharide (LPS), which activates TLRs as well as induces TNF-α production [116]. Once activated, classically activated macrophages have the capacity, via the production of nitric oxide (NO), to eradicate the remaining intracellular pathogens at the site of inflammation [117].

In contrast, alternatively activated macrophages display anti-inflammatory and more profibrotic properties and are mostly associated with tissue repair and secretion of anti-inflammatory mediators [118]. After being exposed to cytokines, such as IL-4, IL-10, or IL-3, macrophages produce polyamines and proline that induce proliferation and collagen production, respectively [117]. These macrophages demonstrate poor antigen-presenting capacity and promote T regulatory (Treg) development [118]. Moreover, in both activations, the amino acid arginine acts as the essential substrate in the pathway. IFN-γ and lipopolysaccharide (LPS)-induced NO synthase 2 (NOS2) degrades arginine into OH- arginine and then further to NO, which can be determined spectrophotometrically by the Griess assay. Arginine then converts into ornithine and further into polyamines and proline [117].

#### 4.1.1. The RAW 264.7 Murine Cell Line

Macrophage cell lines provide a convenient in vitro model for drug screening, as it enables researchers to obtain large numbers of cells for experiments, thereby reducing the cost and time involved in the testing process. In addition, the reproducibility and consistency of clonal cells make them a reliable tool for drug development and toxicology studies. For these studies, a RAW 264.7 murine cell line as a model for macrophages has proven to be an effective way to determine the anti-inflammatory potential of extracts and compounds and, as used worldwide, allows the comparison of experimental values across multiple laboratories in the world [119]. Macrophages, as attractive targets for inflammation, are potent producers of pro-inflammatory cytokines; thus, they enable us to investigate the changes to their phenotype and cytokine production [120]. Although several permanent murine macrophage-like cell lines, which exhibit macrophage-associated effector functions, have been characterized in the past, murine macrophage cell lines have proven to be useful. They can utilize L-arginine to form NO and its stable derivatives, nitrite (NO_2_^−^ and nitrate (NO_3_^−^). NO quantification by macrophages is reliable, rapid, and inexpensive to perform. Additionally, it is flexible and can be adapted to the needs of an individual research laboratory. RAW cells are relatively homogeneous and easily cultured. They can be directly cultured with supernatants or even prepared a day prior. Its supernatant can be tested immediately or frozen for future studies [121,122].

#### 4.1.2. The J774 Murine Cell Line

J774 macrophages represent another well-established model system in cell biology [123] and are commonly used for drug screening purposes in the field of pharmacology and toxicology. Derived from mice, J774 macrophages are well characterized, well maintained, highly proliferative, and easily accessible, making them ideal for laboratory studies. Like RAW 264.7 cells, J774 macrophages have the potential to express inducible nitric oxide (NO) synthase. J774 macrophages can be stimulated by lipopolysaccharide from *Escherichia coli*, inducing the release of NO, cytokines, and prostaglandin E_2_ [124]. In drug screening, J774 macrophages have been used to evaluate the effects of potential therapeutic compounds on these immune cells. The use of J774 cells allows researchers to study the toxicity and efficacy of new drugs, as well as their impact on cellular processes, such as phagocytosis, cytokine production, and oxidative burst [125,126,127].

Thus, RAW 264.7 and J774 macrophages are valuable resources for drug screening, allowing researchers to study the effects of new drugs (including novel CSAIDS) on immune cells and to gather data that can inform the development of new therapeutic strategies.

### 4.2. Microglial Cells as Screening Tools for Anti-Inflammatory Drugs

Microglia are immunocompetent cells of the central nervous system (CNS) [128], which play an important role in the physical and pathological conditions of the brain [129,130]. Well integrated into the neuronal glial network of the healthy CNS, microglia are distributed in all brain regions with varying densities between 5% in the *corpus callosum* and 12% in the *substantia nigra* [131]. The principal function of microglia is to manage brain homeostasis (housekeeping) [132]. As first responders to infection, inflammatory, and pathophysiological stimuli, microglia react to these conditions by altering motility, phagocytic functions, cytokine release, and the expression of innate and adaptive immune-function molecules. Thus, microglia can transform from a quiescent state to different activation states [132]. The central role of activated microglia is in brain defense, to be specific, as dead cell scavengers and immune effector cells [133]. Microglia can be activated into a variety of states, of which a classically activated state (or M1) and an alternatively activated state (or M2) are best characterized [134]. It can become chronically activated by either a single stimulus, such as LPS and neuron damage, or multiple stimuli resulting in cumulative neuronal loss with time [135]. Microglia activation is generally accompanied by partial withdrawal of processes to the cell body, proliferation and expression, and release of pro-inflammatory cytokines, such as IFN-γ, TNF-α, and IL-6. The recruitment of these cytokines results in microglial activation [134].

While attempting to protect neurons, microglia can accidentally kill them as well. Potential mechanisms through which activated microglia kill neurons include the stimulation of phagocyte NADPH oxidase (PHOX)-producing superoxide, expression of iNOS-producing NO, decreased release of nutritive brain-derived neurotrophic factor (BDNF), and insulin-like growth factor (IGF-1) and release of glutaminase, glutamate, TNF-α, and cathepsin B [134]. Activated microglia or brain inflammation has been long implicated in the pathology of neurodegenerative diseases, such as Alzheimer’s disease (AD), Parkinson’s disease (PD) and multiple sclerosis (MS). Activated microglia are considered to be a potential chronic source of multiple neurotoxic factors, such as NO, TNF-α, IL-1β, and reactive oxygen species (ROS), driving neuronal damage and cell death [135].

Therefore, suppression of microglia-mediated inflammation in neurodegenerative disease therapy is considered promising. Anti-inflammatory drugs exerting neuroprotective effects have been shown to repress microglial activation, thus blocking and reducing symptoms of brain inflammation [13].

Microglial cells express a wide range of pattern recognition receptors in the toll-like receptor (TLR) family. Each TLR recognizes a specific type of pathogen-associated molecular patterns (PAMPs), the interaction of which drives the innate immune responses. One of the most potent stimuli that induce microglial activation is LPS, the ligand for TLR-4, which is expressed by microglial cell lines, such as BV-2 and N11 [136,137,138]. Using microglial cell lines reduces the requirement of constantly maintaining primary microglial preparations derived from experimental animals. The next two sections introduce two major microglial cell lines frequently used in the screening of anti-inflammatory drugs.

#### 4.2.1. The BV-2 Microglial Cell Line

The BV-2 cell line, derived from v-raf/v-myc-immortalized murine neonatal microglia [139], demonstrates phenotypic and functional properties of reactive microglia [140]. For in vitro drug screening purposes, the BV-2 cell line is one of the frequently used cell line alternatives to primary microglia (PM) [138,141,142]. The BV-2 cell line is considered an excellent in vitro model for neuroinflammation studies [141].

Similar to RAW cells, lipopolysaccharide (LPS) activation through TLR-4 has been widely used to study the molecular mechanisms of microglial activation in BV-2 cells. LPS alone or in combination with IFN-γ triggers the release of massive pro-inflammatory mediators, including NO and PGE_2_ and cytokines, including TNF-α and IL-6 [142,143,144,145]. LPS is also a potent activator of nuclear factor kappa-light-chain-enhancer of activated B cells (NF-κB), which results is the phosphorylation of the IκB proteins. It also activates the MAPK signaling (also known as Ras-Ref-MEK-ERK) pathways. Reportedly, MAPKs are involved in LPS-stimulated production of COX-2 and iNOS by controlling NF-κB activation in microglial cells [142].

When stimulated with an exogenous application of murine IL-4, BV-2 cells demonstrate an enhanced alternately activated or M2a phenotype [146]. Activation, induced by IL-4, is associated with the increase in arginase 1 (Arg1) and chitinase 3-like 3 (Ym1). In addition, resolvin D_1_ (RvD_1_), which is a specialized anti-inflammatory and pre-solving mediator, is known to promote IL-4-induced microglia (BV-2 cells) M2a activation by increasing the expression of Arg1 and Ym1. It also enhances activation via STAT6 and PPAR γ signaling pathways [147]. The reactive patterns of BV-2 cells after stimulation with LPS show many similarities to that of PM. LPS-stimulated BV-2 cells demonstrate a normal regulation of iNOS production and functional response to IFN-γ. They can stimulate other glial cells as well as induce the translocation of NF-κβ [141].

In a comparative study between rat primary microglia and BV-2 cell lines, both cell lines expressed the microglial activation marker protein, ionized calcium-binding adapter molecule 1 (IBA-1). Both cell lines responded to Aβ fibrils with an increase in phagocytosis, and the induction of genes upon LPS stimulation was 90% similar in BV-2 cells compared to the PM cell line. Thus, BV-2 cells provide an efficient in vitro model for microglia [148].

#### 4.2.2. The N11 Microglial Cell Line

N11 (sometimes also called N-11) is another well-studied microglial model cell line with constitutive and inducible functional activities [149]. The cells are derived and developed by transforming primary mouse embryonic microglial cell lines with v-myc or v-mil oncogenes of the avian retrovirus MH-2 [148]. N11 cells demonstrate positive results for microglial cell markers, such as Macrophage-1 or MAC-1, FcR, and F4/80 and can express the full set of pro-inflammatory cytokines [150]. Upon stimulation, neuroinflammation is produced by the activated N-11 cells, resulting in the production of several pro-inflammatory mediators, including NO, IL-6, and TNF-α through the activation of the NF-κβ pathway [151]. N11 microglial cells can be activated by either LPS, bacterial cell wall proteoglycans, or advanced glycation end products (AGEs) [152,153]. The pathogenic stimulus LPS, which is a well-characterized ligand for TLR-4, induces reactive phenotypes in microglia associated with morphological changes and the release of cytokines, such as IL-1β, TNF-α, IL-6, and NO [143,154]. AGEs are physiological pro-inflammatory inducers that accumulate on long-lived protein deposits [152,155]. When AGEs bind to their receptors, it leads to the activation of redox-sensitive transcription factors, such as NF-κβ, and thus the expression of pro-inflammatory cytokines, including IL-1, IL-6, and TNF-α [155].

The use of immortalized microglia cell lines, such as BV-2 and N-11, eliminates the burden of laborious primary cultures with a short lifetime and provides robust in vitro models for studying brain inflammation and drug screening.

### 4.3. Pro-Inflammatory Activation Pathways of Macrophages and Microglia

#### 4.3.1. Activation via the IFN-γ Pathway

Interferons (IFNs) are proteins with antiviral, antiproliferative, and immunomodulatory effects, which are released from cells in response to a variety of stimuli. IFNs are classified into two types: Type I and Type II. Type I is further divided into seven classes: IFN-α, IFN-β, IFN-ε, IFN-κ, IFN-ω, IFN-δ, and IFN-τ [156]. Of these, IFN-α, IFN-β, IFN-ε, IFN-κ, and IFN-ω exist in humans while IFN-δ and IFN-τ have been described for nonhuman homologs only (pig and cattle) [157].

Type I IFNs influence the development of innate and adaptive immune responses by exhibiting three major functions. Firstly, they induce cell intrinsic antimicrobial states in both infected and neighboring cells, thus restricting the spread of infectious agents, such as viral pathogens. Secondly, Type I IFNs modulate innate immune responses in a balanced manner, thus promoting antigen presentation and natural killer cell functions. At the same time, they prevent pro-inflammatory pathways and cytokine production. Lastly, Type I IFNs activate the adaptive immune system [158].

IFN responses associated with all Type I IFNs are mediated by binding to a common cell surface receptor, known as the Type I IFN receptor. Type I IFN receptor comprises two subunits referred to as IFNAR1 and IFNAR2, which are associated with the Janus kinases-activated (JAKs) tyrosine kinase 2 (TYK2) and JAK1 pathway, respectively [157].

Type II consists only of IFN-γ [156], which is the principal Th1 effector cytokine [159,160].

The production of IFN-γ is mainly regulated by natural killer (NK) and natural killer (NKT) T cells during innate immunity and CD4^+^ or CD8^+^ T cells during adaptive immune response [161,162]. IFN-γ demonstrates antiviral, antitumor, and immunomodulatory functions, thus coordinating both innate and adaptive immune responses [162].

One of the key biological activities of IFN-γ is to activate the macrophages and microglia. It also upregulates a variety of pro-inflammatory mediators, including IL-12, IL-5, TNF-α, interferon-inducible protein-10, iNOS, and caspase-1. Under certain conditions, IFN-γ is also capable of enhancing the activation of the nuclear factor NF-κβ [161].

The canonical IFN-γ-activated pathway proceeds through the Janus kinase (JAK)-signal transducer and activator of the transcription (STAT) pathway [163]. IFN-γ activates its receptor composed of two subunits, IFNGR_1_ and IFNGR_2_ which are intracellularly associated with kinases JAK_1_ and JAK_2_, respectively [162]. The interaction between IFN-γ and IFNGR activates JAKs leading to the phosphorylation, activation, and dimerization of STAT_1_, the major STAT protein activated by IFN-γ. Then, newly formed STAT_1_ homodimers precede its nuclear translocation by annealing to the gamma-activated sequences called an IFN-γ-activated site (GAS), which comprises short DNA elements that bridge STAT binding [159,162,163]. This initiates the transcription of numerous genes; however, the complete transcriptional capacity of STAT_1_ homodimers is achieved after interactions with co-activator proteins, such as p300 and cAMP [162]. As a key player in driving cellular immunity [164], IFN-γ increases the efficiency of the immune system. However, the overactivity of IFN-γ, which causes excessive tissue damage, necrosis, and inflammation has been identified as a potential prerequisite in inflammatory and autoimmune disease pathologies [161,164].

#### 4.3.2. Activation of Cells via the LPS Pathway

Lipopolysaccharide (LPS) is the principal structural component of the outer membrane of Gram-negative bacteria and is one of the best-studied immunostimulatory components for inducing systematic inflammation and sepsis [165]. Exposure of macrophages to LPS essentially releases pro-inflammatory cytokines, which in turn activates further inflammatory cascades, including cytokines, lipid mediators, and adhesion molecules, such as NO, PGE2, TNF-α, ROS, and iNOS. Activation of macrophages via LPS occurs through multiple signaling pathways [166]. The simulation takes place through a series of interactions with various proteins, such as the LPS binding protein (LBP), CD14, MD-2, and TLR4. LBP, a soluble protein, directly binds to LPS, thus facilitating the association of LPS with CD14. Then, CD14, which is a glycosylphosphatidylinositol-anchored protein, facilitates the transfer of LPS to the TLR-4/MD-2 receptor complex exerting LPS recognition [165,167,168].

Upon LPS recognition and stimulation, several intracellular signaling pathways become activated. This includes the IkappaB kinase (IKK)-NF-κB pathway and three mitogen-activated protein kinase (MAPK) pathways. The signaling pathways result in the activation of transcription factors, such as NF-κB and AP-1 [169]. The MAPK pathway is classified into three components: extracellular signal-regulated kinases1/2 (ERK 1/2), c-Jun N-terminal kinase (JNK), and p38 MAPK pathways [166]. MAPKs modulate the functional responses of cells upon stimulation via the phosphorylation of transcription factors and kinases [170]. The (IKK)-NF-κB pathway is known for the rapidity of activation and its unique regulation. In this pathway, inactive NF-κB is retained in the cytoplasm via interactions with inhibitory proteins, the IκBs. The proteolytic degradation of IκB by IKK phosphorylation releases NF-κB into the cytoplasm that in turn activates the NF-κB-regulated target genes. Since NF-κB regulates the transcription of a large number of genes involved in inflammatory responses, the role of IKK-NF-κB signaling is considered to play a major role in inflammatory diseases [165,171].

### 4.4. Pro-Inflammatory Readouts

The discussions hereafter will focus on some of the cellular readouts of inflammation. It will emphasize the biological readouts our group has used in biological assays: NO quantification by the Griess assay and TNF-α by sandwich ELISA, as well as cytotoxicity assays were all employed in the process to isolate novel anti-inflammatory compounds from Australian rainforest plants.

#### 4.4.1. Nitric Oxide

Nitric oxide (NO) is an intercellular messenger molecule involved in the regulation of vascular tone, platelet activation, neurotransmission, inflammation, and host defense mechanisms [172,173,174]. However, when synthesized at higher concentrations by macrophages by iNOS, it becomes cytotoxic to bacteria, viruses, and tumor cells. This is an important mechanism in the host defense; however, as a toxic free radical, NO leads to substantial tissue destruction, especially in the brain [172,174,175]. It is produced in large amounts by immune and nonimmune cells upon induction by mediators, including cytokines during inflammatory responses [176]. NO is synthesized by the enzymatic activity of nitric oxide synthase (NOS) via the L-arginine-NO pathway during which NOS catalyzes the reaction of arginine with molecular oxygen forming citrulline and NO [172]. Among the three NOSs, inducible NOS (iNOS) is involved in the overproduction of NO and is expressed in response to IL-1β, TNF-α, and LPS, the genetic expression of which is commanded by the NF-κβ macrophages [177]. It is understood that secondary metabolites capable of inhibiting inducible NOS and the induction of NF-κβ activation may be of therapeutic benefit in treating inflammation [177,178]. For in vitro studies, the Griess assay is ideally used to assess NO production in RAW 264.7 cells following treatment with bacterial LPS only or in combination with IFN-γ [179].

Quantification of NO in biological matrixes requires careful consideration as it oxidizes rapidly to nitrite and/or nitrate with its half-life ranging from less than 1 to 30 s. Nitrate and nitrite can be measured directly by UV absorbance measurement, chromatographic, and capillary electrophoresis methods; however, these are expensive and time-consuming procedures. One of the ways to measure integrated NO production is by measuring the concentrations of nitrite and nitrate end products whereby the measurement of total nitrate/nitrite concentration is consistently used as an index of NO production [180]. The Griess assay is widely used to measure NO formation in cell culture supernatants after reaction into nitrite. Murine RAW 264.7 macrophages are seeded and incubated with compounds of interest. Upon activation by LPS and IFN-γ, an unstable NO molecule is released by macrophages, which rapidly converts into nitrite and is detected by the Griess reagent [180]. The released nitrite is first treated with sulfanilamide (SA), a diazotizing agent in an acidic medium, to form a diazonium salt, which reacts with N-naphthyl-ethylenediamine (NED), a coupling reagent to form a stable azo compound [179]. Given the cost-effectiveness and simplicity of the Griess test, it is extensively used in high-throughput screening for identifying the anti-inflammatory potential of bulk crude extracts. Quantification of NO in its nitrite form is indicative of the anti-inflammatory activity of the test compounds. The quantity is expressed as an IC_50_ value, which means the half-maximal inhibitory concentration expressed in μg/mL of the test compound, which inhibits NO production halfway between the concentration of NO in fully activated and nonactivated RAW 264.7 macrophages.

#### 4.4.2. Tumor Necrosis Factor-α

Tumor necrosis factor-α (TNF-α) is a pro-inflammatory cytokine that plays a key role in immunity and inflammation. Produced in response to inflammation and other pathological challenges, TNF-α prompts a broad spectrum of cellular responses, including inflammation, cell proliferation, differentiation, and apoptosis [181]. TNF-α exerts its effects by binding to its two distinct receptors: TNFR1 and TNFR2, which results in the recruitment of signal transducers that activates effectors. Through signaling and networking, these effectors lead to the activation of caspases and transcription factors, including activator protein 1 (AP-1) and NF-Κb. AP-1 and NF-Κb are known for inducing genes that participate in acute and chronic inflammatory responses [181].

Additionally, it is known to participate in leukocyte adhesion to epithelium via the expression of adhesion molecules, thus leading to leukocyte accumulation, adherence, and migration from capillaries into the brain [182,183]. Since TNF-α plays a critical pathogenetic role in inflammatory diseases, its antagonists are considered effective in the treatment of these diseases [184]. Recombinant proteins, which act as TNF-α antagonists are an emerging class of therapeutic agents, and in particular, antibodies that neutralize TNF-α have been pursued as potential inhibitors. Inhibition of TNF-α is a proven effective therapy for patients with rheumatoid arthritis, psoriasis, and inflammatory bowel diseases [184].

The TNF-α concentration can be determined using an enzyme-linked immunosorbent assay (ELISA) [74,100,185], which is a solid-phase sandwich ELISA designed to detect and quantify the level of TNF-α in human serum, plasma, or cell culture supernatants. The assay is based on a solid-phase enzyme immunoassay with horseradish peroxidase (HRP) as the marker enzyme. The protein, HRP is employed to catalyze the oxidation of substrates by hydrogen peroxide (H_2_O_2_) to form a fluorescent product in the presence of TNF-α. ELISA kits are commercially available and thus suitable for high throughput screenings. The process is time-consuming; however, it is easy to follow and perform.

### 4.5. Cytotoxicity Assays

#### 4.5.1. The Alamar Blue (Resazurin) Assay

Preclinical toxicity studies for newly discovered compounds performed during the early discovery process have been deemed a promising strategy to exclude toxic drugs from further evaluation to avoid costly animal toxicity studies [186]. The two major parameters of cell viability, which can be easily tested in 96-well formats, are either based on mitochondrial activity or membrane integrity.

The Alamar Blue (resazurin) assay is a widely used assay for assessing cell viability but is also used to measure cell proliferation. The assay is based on the principle that metabolically active cells can reduce the blue dye resazurin to its fluorescent pink product, resorufin, which can be measured using a fluorescent plate reader. Resazurin is a nontoxic, blue dye that is taken up by cells via passive diffusion. Once inside the cell, the dye is reduced by mitochondrial enzymes to its fluorescent product, resorufin. The reduction in resazurin to resorufin requires the transfer of electrons, which is an indicator of metabolic activity in the cells. The reduction in resazurin to resorufin generates a change in fluorescence intensity that can be measured using a fluorescence plate reader [187]. The fluorescence intensity is proportional to the number of viable cells present, as more viable cells will reduce more resazurin to resorufin. The Alamar Blue assay is a sensitive and nondestructive method for monitoring cell viability over time, making it an ideal tool for high-throughput drug-screening discovery studies. Resazurin is extremely stable, water-soluble, economical, ready to use, and minimally toxic to cells [188]. For the benefit of screening large chemical extract libraries, the Alamar Blue assay is also cost-effective when prepared in the lab from the dye resazurin.

#### 4.5.2. The MTT Assay

The MTT (3-[4,5-dimethylthiazol-2-yl]-2,5 diphenyltetrazolium bromide) assay is another well-characterized and simple to perform colorimetric cell viability assay [189]. The assay is based on the ability of live cells to convert the yellow tetrazolium salt MTT (3-(4,5-dimethylthiazol-2-yl)-2,5-diphenyltetrazolium bromide) into a purple formazan product by mitochondrial dehydrogenase enzymes. In contrast to Alamar blue, the produced purple formazan crystals are not soluble in aqueous solutions, and the cells are treated with a solubilization buffer that dissolves the formazan crystals and creates a colored solution. The solution is then quantified by light absorbance at the specific wavelength of 492 nm [190,191]. The MTT assay is one of the preferred cell viability assays as it is cost-effective, easy, and safe to conduct and has high reproducibility. However, formazan crystals are insoluble in water; therefore, before measuring absorbance, an organic solvent, such as DMSO or isopropanol, is required to solubilize them, which destroys the cells [192].

#### 4.5.3. The Trypan Blue Assay

Another simple and widely used method to measure the number and proportion of viable cells in a cell population is the trypan blue dye exclusion method. This method is based on the principle that viable cells possess intact cell membranes that exclude specific dyes whereas nonviable cells do not [192,193].

If a cell suspension is subjected to trypan blue, the viable cells will exclude the dye, which is visually indicated by a clear cytoplasm whereas nonviable cells will show a blue cytoplasm indicating that they did not exclude the dye [194]. Trypan blue is the most extensively used dye, but there are others, including erythrosine B, Congo red, and eosin [192,195]. This method is inexpensive and is a good indicator of membrane integrity indicating the number of living cells. One of the disadvantages is that cells need to be counted manually, unless specialized microscopic and counting software is used. This leads to possible accounting errors and logistic difficulties when processing large sample numbers.

#### 4.5.4. Other Cytotoxicity Assays

There are several more toxicity assays used to determine cell viability.

The LDH assay: This assay measures the release of lactate dehydrogenase (LDH) into the cell culture medium as a result of cell membrane damage. The assay is based on the principle that LDH is a cytoplasmic enzyme that is released into the medium when cells are damaged [196].

The ATP assay: This assay measures the amount of adenosine triphosphate (ATP) present in cells. ATP is a marker of viable cells as it is rapidly degraded after cell death. The assay is based on the principle that luciferase catalyzes the oxidation of luciferin in the presence of ATP, producing light that is proportional to the amount of ATP present [197].

The neutral red assay: This assay is used to assess the viability of cells in culture by measuring the ability of cells to uptake and retain the neutral red dye. The dye is taken up by viable cells and accumulates in the lysosomes. The amount of dye accumulated by the cells is proportional to the number of viable cells. After a specified period of incubation, the cells are washed to remove any unbound dye and fixed with a solution, such as formalin. The dye that has accumulated in the lysosomes is then solubilized with a solution, such as acetic acid, and the amount of dye present is measured using a spectrophotometer [198].

## 5. The Triculture Model for Initial Drug Screening Targeting Neuroinflammation

Neuronal cocultures refer to in vitro systems where neurons are cultured with other cell types, such as microglia. The most used cell culture system to investigate anti-inflammatory and neuroprotective compounds are neuron-microglia cocultures. In this type of coculture, neurons and microglia (primary or cell lines) are cultured together. This can be used to study the interaction between microglia and neurons, as well as the role of microglia in neuronal development and function. We have previously used this system, with a combination of N11 microglia and GFP-transfected Neuro2a cells to determine the anti-inflammatory and neuroprotective effects of selected herbal compounds [199].

The most advanced “brain in a dish” system, which also includes a model of the blood–brain barrier, involves the use of transwell plates. Neuroinflammation involves all the cells present within the neurovascular unit (NVU) and their communication, including the neurons, brain endothelial cells, and microglia [200]. The activation of glial cells (mainly microglia) and consequential expression of pro-inflammatory mediators, such as NO, TNF-α, and IL-6, are shown to impair the blood–brain barrier (BBB) and result in neuron damage [201]. Thus, robust in vitro and in vivo models that can measure the multifaceted interactions of cells within the neurovascular unit (NVU) are key to understanding the complex pathophysiology of neuroinflammation and developing therapeutic interventions. In vitro two-dimensional (2D) or three-dimensional (3D) coculture models are the standard tool to understand and analyze the molecular mechanisms underlying neuroinflammation. In particular, the 2D co- or triculture model shows the advantages that they are simple, fast, cheap, and relatively easy to establish. We have established a 2D triculture model (microglial, endothelial, and neuronal cells) with a simple transwell system in a six-well plate to simulate the NVU environment under neuroinflammation. This triculture model incorporates three different cells that play important roles in the pathological events of neuroinflammation, microglia N11 (N11), brain endothelial MVEC(B3) (MVEC), and neuronal N2A (N2A) cells. The MVEC cells are growing on the flat surface of the transwell to form the endothelial tight junction, while the microglia N11 cells are seeded at the bottom of the transwell after the polylysine coating. The transwell is then placed inside a flat-bottom six-well plate with N2A cells. Transwells have been extensively used that consist of a luminal and a basal compartment separated by a microporous membrane that allows the transfer of molecules from one compartment to the other. In this system, the transwell allows the communications among N11, MVEC, and N2A to grow in different compartments. LPS is used to trigger the activation of N11 cells in the triculture system, which induced the overexpression of NO, TNF-α, and IL-6 mediated by the upregulated TLR4-NF-κB signaling pathway. The overproduction of pro-inflammatory mediators is shown to penetrate the transwell and damage the endothelial tight junction of MVEC cells as evidenced by higher permeability and reduced protein expression of zonula occludens-1 [151].

This triculture neuroinflammation model mimics the microenvironment, the cellular crosstalk, and the molecular events among microglial, endothelial, and neuronal cells that take place during neuroinflammation. This model has been cited for its potential applications for neurological disorders, such as AD. In addition, it provides a robust in vitro model for screening potential therapeutics to treat various neurodegenerative diseases before animal testing [151].

## 6. Animal Models of Neuroinflammation

To validate the potency of anti-inflammatory natural compounds and extract, there is a need to carry out preclinical studies in appropriate animal models before the compound can move further to clinical trials. While there is a lack of consistency between the models, we have listed some of the more frequently used models that sufficiently mimic both acute and chronic inflammation accompanying human diseases.

### 6.1. Rodent Models of Acute Neuroinflammation

One of the most common in vivo and in vitro models of neuroinflammation is induced by lipopolysaccharides (LPS), which are large molecules consisting of a lipid and a polysaccharide that are bacterial toxins. LPS-induced neuroinflammation models are more relevant in mimicking the acute inflammatory response of glial cells. LPS binds CD14 on microglia membranes when applied locally. To activate the microglia, the LPS-CD14 complex then interacts with the TLR-4 on the membrane. In turn, this will set off signal transduction cascades that will encourage the quick transcription and release of chemokines (CCL2, CCL5, and CXCL8), complement system proteins (C3, C3a, and C5a receptors), pro-inflammatory cytokines (IL-1, IL-6, IL-12, p40, and TNF-), and anti-inflammatory cytokines (IL-10 and TGF-β). There are variations in LPS-induced neuroinflammation models depending on the delivery method, exposure duration, and age of the animal [202].

A study using a primary neuron–glia culture from rat midbrain reported that LPS is capable of inducing neurotoxicity, microglial activation, and pro-inflammatory cytokine release, as well as neurodegeneration [203]. An in vitro study demonstrated the direct effect of acute inflammation on neuronal circuits using rat acute hippocampal slices exposed to low levels of LPS (10 μg/mL) for 30 min [204]. This study reported a significant increase in the TNF-α and IL-1β concentrations, as well as an increase in the epileptiform discharge frequency and spikes per burst [204]. A similar in vitro cell culture study on mice cortical network (neuron/astrocyte/microglial) demonstrated comparable results [205]. The same study, with the use of confocal analysis of immunofluorescence, reported numerous GFAP-positive astroglia surrounding this cortical neuronal network. Furthermore, they observed both ramified/resting and round amoeboid/activated LEA lectin^+^ microglial cells in control cultures (i.e., not LPS treated). While in the LPS-treated cultures, there was a higher number of LEA lectin^+^ microglia arranged in clusters with predominant round morphology [205].

After systemic (i.p.) LPS-injection, normal-aged mice (BALB/c mice, 18- to 20-month-old) presented an amplified neuroinflammatory response, prolonged sickness/depressive-like behavior, and impaired working memory [206,207,208,209,210]. Moreover, aged mice displayed a hyperactive microglia response associated with increased induction of inflammatory IL-1β and anti-inflammatory IL-10 in the brain [210]. The same study also reported, in aged mice, a higher TLR-2 expression level and prominent induction of IL-1β in MHC II^+^ microglia. An in vivo study, after a single intrahippocampal LPS injection into mice, reported an inflammatory response that lasted at least 3 weeks, including an acute transient pro-inflammatory cytokine release as well as a subacute and sustained change in microglial morphology [211]. However, it did not present significant astrogliosis or cell death. The same study also reported a rapid increase in the expression of IL-1β and IL-6 at 12 h, which then returned to either low or undetectable levels at 3 days and 3 weeks after LPS treatment [211]. Another study investigated the acute effect of systemic LPS administration in different mouse brain regions (frontal cortex, parietal cortex, hippocampus, striatum, and cerebellum), and the different expression levels of cytokines and microglial activation in these brain regions [212]. In this study, LPS-injected mice displayed higher levels of IL-1β mRNA in the frontal cortex, parietal cortex, hippocampus, and striatum compared to saline controls [212]. Moreover, the Iba1^+^ cell number in the striatum, medial septum, frontal cortex, dentate gyrus, and the CA1 and CA3 hippocampal regions was higher in the LPS-treated compared to saline-treated mice. This indicated significant neuroinflammation in LPS-treated mice. However, the LPS-treated mice showed no considerable changes in spatial memory, despite a significant neuroinflammatory response [212]. LPS-induced neuroinflammation has increased Aβ burden, impaired cognitive function, and induced neurodegeneration [213,214,215].

A recent study demonstrated the age-related susceptibility of medial septal cholinergic neurons to glial activation mediated via a peripheral LPS (500 μg/kg) injection to transgenic ChAT^(BAC)^-eGFP mice at different age groups: young (3–6 months), adult (9–12 months), and aged (18–22 months) [216]. In this study, TNFα levels increased significantly in all age groups confirming the activation of microglia and the existence of acute neuroinflammation.

### 6.2. Rodent Models of Chronic Neuroinflammation

Chronic neuroinflammation is a noxious inflammatory response characterized by an increased presence of activated microglia, astrocytes, complement proteins, cytokines, and reactive oxygen, nitrogen, and carbonyl species. It increases with aging and is a key feature in neurodegenerative diseases, such as AD [217]. In experimental settings, chronic neuroinflammation can be studied using several rodent models: immune challenged-based, toxin-induced, and (non-AD) transgenic models [202].

#### 6.2.1. Immune Challenged-Based Models

A recent study demonstrated that chronic induction of LPS into the lateral ventricle of a rat brain caused inflammatory reactions and memory impairment in the rats [218]. They also reported that LPS infiltration significantly increased TNF-α, IL-6, and IL-1β levels in the hippocampus, ultimately leading to a chronic neuroinflammatory response in the brain, further accentuated by increased NF-κB levels and increased mRNA expression levels of iNOS, TLR4, and BDNF [218]. Chronic LPS injection into C57BL/6J mice (both i.p. and i.c.v) induced microglia (Iba-1 labeled) activation through a NF-κB signaling pathway and augmented the levels of pro-inflammatory molecules (including TNF-α, IL-1β, PGE2, and NO) while attenuating the level of anti-inflammatory molecules (including IL-4 and IL-10) [219]. Furthermore, LPS-induced systemic inflammation and neuroinflammation have been shown to elevate Aβ levels and neuronal cell death resulting in sickness behavior and cognitive impairments associated with neurodegenerative conditions, such as AD [219]. Several other studies have also confirmed that chronic LPS administration increases Aβ deposition and leads to spatial memory and learning impairments, similar to those witnessed in AD [202,217,220].

It is well understood that a peripheral immune insult can challenge the innate immunity of the brain through microglial activation and then trigger neurodegenerative processes [206,208]. This is especially problematic in an aging brain when neuroinflammation becomes chronic [31]. Studies on animal models have shown that maternal systemic infection during gestation can increase the risk of cognitive abnormalities in the offspring [202]. The viral mimic, polyriboinosinic-polyribocytidilic acid (PolyI:C), is a synthetic analog of double-stranded RNA, which is used to stimulate the immune system of experimental animals [221]. These models are used to study the effects of life-long neuroinflammation on cognitive function [221]. Studies using mice models of a PolyI: C-induced systemic immune challenge have reported inflammation in the injected pregnant animal and a chronic pro-inflammatory state in the fetus [221,222]. Krstic et al. (2012) reported that a systemic PolyI:C injection on late gestational day 17 induces chronic neuroinflammation with significant augmented levels of hippocampal cytokines, including IL-1β, IL-6, and IL-1α [222]. Compared to the control group, these changes were observed as early as the age of 3 weeks and sustained throughout aging [222]. They also reported, in double immune-challenged mice, significantly higher levels of tau hyperphosphorylation at 6 and 15 months of age. As well as initiation of amyloidogenesis at late as 12 months of age compared to controls [222]. Moreover, at 20 months of age, the immune-challenged animals displayed significant impairments of spatial recognition memory compared to the control group [222]. The microglia activation in double immune-challenged mice was also accompanied by astrogliosis [222,223], suggesting that a prenatal immune challenge induces subacute microglia activation and increases inflammatory cytokine levels in adulthood and then gradually deteriorates the brain’s innate immune system with aging [222]. The second immune challenge accompanied by astrogliosis further accelerates this process, leading to dysregulated brain metabolism and neuronal health [222,224,225]. Evidence suggests that in the CNS the immune challenge by PolyI:C is mainly mediated by TLR-3-induced activation of microglia followed by an NF-kB-dependent pro-inflammatory cytokine induction (including IL-1β, IL-6, IL-8, TNF-α, and type I and II interferons) [202,221,226].

#### 6.2.2. Toxin-Induced Inflammation Models

Streptozotocin (STZ) is a glucosamine–nitrosourea compound that selectively damages pancreatic β-cells once taken up by the glucose transporter [202]. It is a peripheral toxin that is used to induce diabetes mellitus in animal models. Peripheral (i.v) STZ-injected diabetic rats have shown brain aging related to AD-like pathology [227]. These animals, 4 months after toxin induction, displayed frontal lobe neurodegeneration, hippocampal atrophy, Aβ aggregation, and synapse loss, followed by cognitive impairments [227]. A single STZ (1 or 3 mg/kg) i.c.v injection in rats has presented with chronic neuroinflammation, followed by atrophy and severe neuronal cell loss in the septum and corpus callosum (a more pronounced effect with 3 mg/kg) [228].

Colchicine is a poisonous plant alkaloid derived from *Colchicum autumnale* seeds [229]. It irreversibly binds to tubulin molecules and disrupts microtubule polymerization-blocking axonal transport [229]. This will then jeopardize normal neuronal function and lead to neurodegeneration [202,229]. A single i.c.v injection of colchicine in rats has been shown to cause AD-like neurodegeneration and neuroinflammation in different brain areas [229]. Impaired memory was reported along with neurodegeneration accompanied by a significant presence of plaques. Nissl granule chromatolysis in several brain areas (frontal cortex, amygdala, parietal cortex, corpus striatum) was evident, with the greatest severity reported in the hippocampus [229]. Moreover, hippocampal IL-1β, TNFα, ROS, and nitrite levels were significantly increased [229].

#### 6.2.3. Transgenic Inflammatory Mouse Models

The pro-inflammatory cytokine IL-1β-driven neuroinflammation is thought to contribute to the disease pathology of chronic neurodegenerative diseases, such as AD. To better define the role of IL-1β in neuroinflammation and AD, Shaftel et al. (2007) engineered the IL-1β^XAT^ mouse model [230]. In this model, after transgene activation, IL-1β is continuously overexpressed with robust neuroinflammation, which lasts for months [230]. Chronic robust neuroinflammation was evident in the IL-1β^XAT^ mice hippocampus, depicted by an increased number of amoeboid-shaped activated microglia co-expressing Iba-1 and MHC class II, compared to wild-type [230]. The same study reported elevated levels of GFAP attributing to chronic astrocyte activation along with increased expression of all members of the murine IL-1 (mIL-1) family pro-inflammatory cytokines, as well as TNF-α and IL-6. Surprisingly, robust IL-1β overexpression within the hippocampus of the APP/PS1 mouse model of AD led to a reduction in amyloid pathology [230]. Explaining a possible adaptive role of IL-1β by enhancing microglial activation and phagocytosis of plaque [230]. It is noteworthy that depletion of cortical cholinergic innervation is a well-known pathological characteristic of normal aging and AD, and both short-term and long-term IL-1β overexpression was unable to deplete cholinergic fibers in APP/PS1 mouse model [231,232,233].

Cell cycle regulation is dependent on cyclins and cyclin-dependent kinases (Cdks). Among the Cdks, Cdk5 is particularly involved in brain development by promoting neuronal growth [202]. The Cdk5 is activated by a regulatory subunit p35. However, p35 can be cleaved into a smaller fragment p25 by calpain (calcium-activated protease) [234]. The Cdk5/p25 is suggested to hyperactivate Cdk5 leading to tau hyperphosphorylation, a precondition for neurofibrillary tangle (NFT) formation [234]. The generation of truncated protein p25 contributed to toxicity in AD [235]. This motivated the generation of a transgenic mouse model overexpressing p25 [235]. Transgenic mice overexpressing human p25 have been shown to induce neuroinflammation in vitro and in vivo but in the absence of any amyloid or tau pathology [234]. The same study revealed that p25 overexpressing neurons release a soluble lipid factor, lysophosphatidylcholine (LPC), which causes astrogliosis and increased pro-inflammatory cytokine production, leading to subsequent neurodegeneration. They further reported a marked increase in GFAP and pro-inflammatory cytokines in p25Tg mice brains, even after 1 week of induction. However, microgliosis was absent at this time point and was apparent only during later induction periods [234]. It seems that p25 levels are specifically reduced in the early stages of AD, and p25 generation/overexpression in the mouse hippocampus is associated with normal memory formation and improved synaptogenesis [236]. Furthermore, a transient increase in p25 levels in the hippocampus enhanced long-term potentiation (LTP) and facilitated hippocampus-dependent memory [237]. However, prolonged p25 production led to severe cognitive deficits accompanied by synaptic and neuronal loss, as well as impaired LTP [237].

Cytokines are important soluble regulatory proteins released between cells during both physiological and pathological states [238]. Within the CNS, some cytokines are implicated in the regulation of sleep, long-term memory, attention, and BBB integrity, as well as in neurodegenerative diseases [238]. IL-6 is a typical cytokine found in low levels under normal physiological conditions in the brain and has a wide variety of biological actions that overlap with other cytokines, such as IL-lα/, β, and TNF-α [238]. The localized IL-6 production in the CNS is not only mediated by infiltrating immuno-inflammatory cells but also by astrocytes and microglia [239]. A dramatic surge in IL-6 expression and secretion is observed in various neurological disorders, including, AD, PD, MS, bacterial meningitis, and brain trauma [238]. The GFAP IL-6 transgenic mouse model was generated as a chronic neuroinflammation model to investigate cytokine signaling within the CNS [239]. In this model, astroglia expresses the IL-6 gene under the transcriptional control of the glial fibrillary acidic protein (GFAP) promoter, ensuing brain-specific overexpression of IL-6 [217]. The IL-6 levels are undetectable in the wild-type (WT) mice brains [217], in contrast, transgenic IL-6 mice display significantly high levels of IL-6 in the cerebellum, the striatum, the hippocampus, the hypothalamus, the neocortex, and the pons [240], with approximately 50 times more IL-6 levels detected in the cerebellum compared to WT [241]. The GFAP-IL-6 mice present age-related structural changes and cellular changes starting from 3–6 months of age, including microglia and astrocyte activation, lowered hippocampal neurogenesis, and neurodegeneration accompanied by age-related deficits in learning and memory [217,240]. It is clear that in many respects this transgenic mouse model replicates neuropathological changes in human neurodegenerative diseases, including AD and HIV-associated dementia [217]. The homozygous GFAP-IL6 mice develop phenotypically aggressive pathology comprised of tremors, ataxia, and seizures, and they also tend to die prematurely [242]. Whereas the heterozygous GFAP-IL6 mice develop milder pathology, including astrocytosis, microgliosis, angiogenesis, and neurodegeneration, and they do not die prematurely [242,243]. Early studies have shown that neurodegenerative changes and progressive learning deficits in GFAP-IL6 mice are correlated with the degree of microgliosis [244,245]. A recent study reported elevated Iba1^+^ microglia numbers in the cerebellum of GFAP-IL-6 mice as early as at 3 months of age, as well as at all time points (3,6, 14, and 24 months of age) compared to WT (C57BL/6) mice [242]. Moreover, TNF-α levels were significantly higher in the GFAP-IL6 compared to WT mice at all time points [242]. Additionally, GFAP-IL6 mice reported cerebellar volume loss from the age of 6 months, as well as loss of fine and gross motor coordination that worsened with age [242]. Compared to nondemented elderly people, IL-6 can be consistently detected in the brains of AD patients [217]. Thus, elevated serum IL-6 levels can be used to differentiate dementia from normal aging [217]. Based on previous studies and preliminary data, GFAP-IL6 mice models seem to be a promising transgenic rodent model to study the detrimental effects of chronic neuroinflammation on the mammalian brain, and special attention should be driven toward its effects on the cholinergic system.

#### 6.2.4. Validation of the Anti-Inflammatory Effect of Curcumin in the GFAP-Il6 Mouse, a Transgenic Inflammatory Mouse Model

Although not a compound isolated from Australian Bush medicine, the anti-inflammatory properties of another plant-derived compound, curcumin, were tested in the GFAP-IL6 mouse. Curcumin, which is extracted from the spice turmeric (*Curcuma longa*), has numerous pharmacological effects (antioxidant, anti-inflammatory) suggesting a therapeutical potential for AD and other neurological disorders [246,247]. Curcumin penetrates the BBB barrier, and it is safe in humans and rodents [248,249,250].

A crucial characteristic of the anti-inflammatory effect is the downregulation of COX-2, iNOS, TNF-α, and several interleukins, which especially inhibits STAT3 phosphorylation and the consequent nuclear translocation mediated by IL6 and also blocks the downstream signaling initiated by the IL6 receptor during microglia activation [9,62]. Despite its multi-neuroprotective properties, curcumin has several pharmacokinetic limits, such as poor absorption, rapid metabolism, and elimination (Figure 2) [251].

To enhance the bioavailability of curcumin, alternatives, such as structural analogs of curcumin or the use of adjuvants, such as piperine and phospholipids, have been tried. To improve its bioavailability, a nanoparticle-based delivery of curcumin is the best approach to increase its bioavailability [251]. To counteract the low bioavailability of such phytoconstituents, curcumin is incorporated into a lipid-compatible molecular complex by using small lipid molecules, such as phosphatidylcholine, due to its high capacity to cross the lipid-rich biomembranes of the GI tract and to reach the circulation. Indeed, curcumin formulations encapsulated in lipid-based matrices (liposomes and micelles) have been developed to improve the bioavailability of curcumin to cross the BBB and reach the brain. Examples are Longvida^®^ Optimized Curcumin (Verdure Sciences Inc., Noblesville, IN, USA) and Meriva^®^ (Indena), which can achieve therapeutically relevant concentrations in the brain in the range of μM concentrations in the rodent brain [63,252].

Our work has demonstrated a significant reduction in the number of activated microglia and astrocytes in the hippocampus and cerebellum in a study involving Longvida^®^ Optimized Curcumin administered to GFAP-IL6 mice [253,254]. In GFAP-IL6 mice, feeding with liposomal curcumin (Meriva) from 2 months to 6 months of age reduced (1) the number of Iba1+ in the hippocampus and cerebellum, (2) the number of TSPO+ cells in the cerebellum, and (3) the number of GFAP+ cells in the hippocampus [253]. Curcumin feeding also increased synaptophysin levels and improved motor performances in the GFAP-IL6 mice [253]. These pieces of evidence suggest that curcumin formulations potentially reverse the detrimental effects of chronic glial activation during neuroinflammation and could be used as a treatment for neurodegenerative diseases. These in vivo studies strongly suggest that curcumin can interfere with a broad range of neuroinflammatory processes and can inhibit glial cell activation.

## 7. Conclusions and Future Studies

The Australian bush is a rich source of medicinal plants that have evolved in the face of dramatic environmental challenges due to its prolonged geographical isolation. This review has illustrated various examples of how the search for potent anti-inflammatory compounds from plant source pathways from the discovery of bioactive plants through to the structural identification of active compounds, testing for potency, and then neuroprotection in a triculture system, and finally, the validation in the GFAP-IL6 mouse strain similar to the curcumin feeding experiments. Future studies will include the testing of the identified extracts and compounds “from the bush” in this mouse model of neuroinflammation, before the initiation of toxicology studies and the transition to humans.

## Figures and Tables

**Figure 1 ijms-24-11086-f001:**
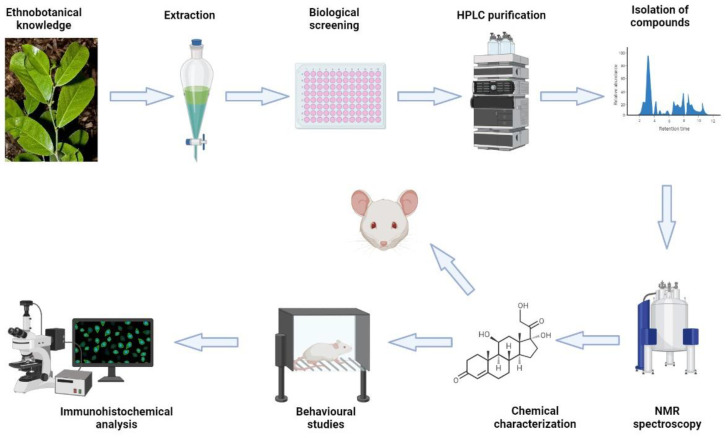
Drug discovery pipeline for herbal compounds. The discovery of bioactive plants is performed through traditional use in medicine and ethnobotanical studies, followed by high-throughput screening of plant extracts. Once bioactive plants are identified, the active compounds responsible for the observed effects are isolated and purified. This can be achieved using various chromatographic techniques. The structure of the active compound(s) is determined using techniques, such as nuclear magnetic resonance (NMR) spectroscopy, mass spectrometry, and X-ray crystallography. The purified compounds are then tested for their potency using in vitro and/or in vivo assays. In vitro assays involve testing the compounds on isolated cells or tissues, while in vivo assays involve testing the compounds in animal models. Once the compounds are potent, their neuroprotective effects can be studied in a triculture system, consisting of neurons, astrocytes (or endothelial cells for BB penetration), and microglia. This system mimics the complex interactions that occur in the brain and provides a more accurate representation of the neuroprotective effects of the compounds. Finally, the compounds can be validated in appropriate neuro-inflammatory mouse models. These models are used to study neuroinflammation, a key factor in many neurodegenerative diseases. The compounds are tested for their ability to protect against neuroinflammation and to prevent or slow down the progression of neurodegenerative diseases.

**Figure 2 ijms-24-11086-f002:**
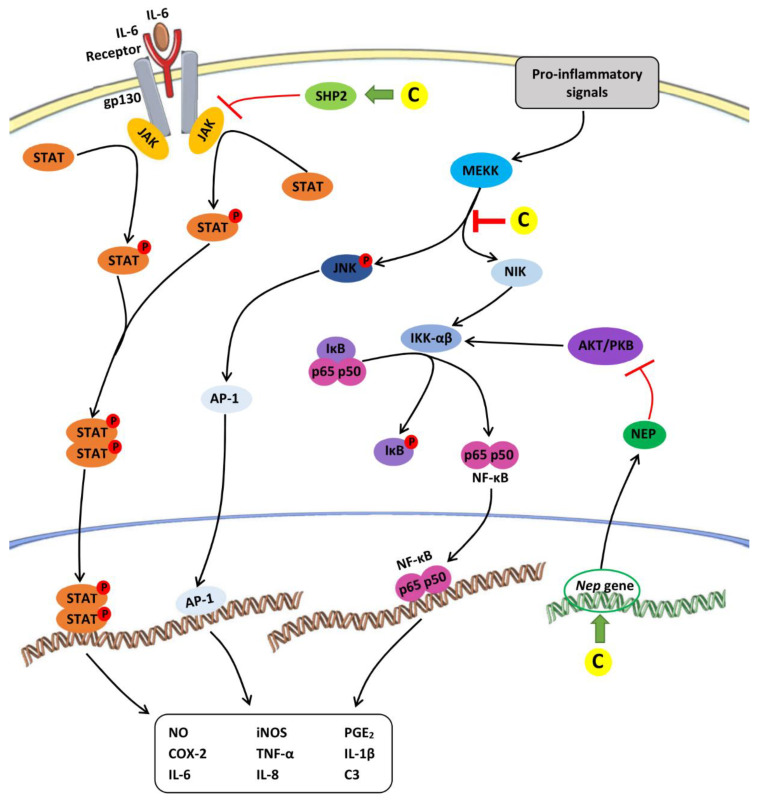
The pro-inflammatory IL-6 cascade and the actions of curcumin. Curcumin (C) can interfere with the Janus kinase/signal transducer and activator of transcription (JAK/STAT) pathway and decrease STAT phosphorylation by JAK inactivation via the upregulation of Src homology 2 domain-containing protein tyrosine phosphatase (SHP2). It can block the signaling cascade upstream of phosphor-c-jun NH2-terminal kinase (JNK) and nuclear factor κB-induced kinase (NIK) and, as a consequence, inhibit both AP-1 and nuclear factor (NF)-κB nuclear translocation. Curcumin can also promote the transcription of the gene coding for neprilysin (NEP), which inhibits AKT/protein kinase B (PKB) that can no longer activate the NF-κB pathway. By interfering with the pro-inflammatory pathway, curcumin downregulates the transcription of pro-inflammatory factors, such as nitric oxide (NO), inducible nitric oxide synthase (iNOS), prostaglandin (PG)E2, cyclooxygenase (COX)-2, tumor necrosis factor (TNF)-α, interleukin (IL)-1β, IL-6, IL-8, and complement 3 (C3).

**Table 1 ijms-24-11086-t001:** Selected potent anti-inflammatory agents isolated from Australian plants.

Plant Species	Constituent(s) Identified	IC_50_ in μM (NO Inhibition)	IC_50_ in μM (TNF-α Inhibition)	Cytotoxicity/LC_50_ in μM	Therapeutic Index (TI) (Compared to NO)	Reference
*Alphitonia petriei* (Rhamnaceae)	*trans-* coumaroyl ester of alphitolic acid	1.7	10.9	4.8	2.8	[103]
	*cis*-coumaroyl esters of alphitolic acid	3.5	5.6	8.0	2.3	
*Elaeocarpus Eumundi* (Elaeocarpaceae)	New phenolic monosaccharide related to pieceid-2-Ogallate	22.6	39.7	>165.4	>7.3	[104]
	dihydropieceid	73.4	25.6	>230.2	>3.1	
*Eupomatia laurina* (Eupomatiaceae)	eupomatenoid-2	28.2	57.4	21.9	0.8	[105]
	Eupomatene 4	32.7	44.1	38.8	1.2	
*Ternstroemia cherryi* (Theaceae)	Ternstroenol B	0.7	2.1	2.3	3.2	[106]
	Ternstroenol D	0.9	3.8	3.2	3.5	
*Cryptocarya* *mackinnoniana* (Lauraceae)	Cryptocaryoic acid A	18.4	>100	>100	>5.4	[107]
	Cryptocaryoic acid B	27.6	>100	>100	>3.6	
*Tristaniopsis laurina* (Myrtaceae)	Tristaenone A	37.6	80.6	>250	>6.7	[101]
	8-desmethyleucalyptin	16.2	46.6	>250	>15.1	
*Backhousia mytifolia* (Myrtaceae)	Myrtinol A	11.5	24.5	18.8	1.6	[98]
	Myrtinol E	8.5	17.2	15.5	1.8	
*Syncarpia glomulifera* (Myrtaceae)	Sideroxylin	2.3	20.8	19.4	8.4	[108]
	Tetragocarbone C	3.9	16.9	18.4	4.7	
*Waterhousia* *Mulgraveana* (Myrtaceae)	Mulgravanol B	42.0	140.8	202.0	4.8	[109]
	Mulgravanol C	35.2	128.0	135.0	3.8	
*Pleuranthodium racemigerum* (Zingiberaceae)	1-(4″-Methoxyphenyl)-7-(3′,4′-*di*-hydroxyphenyl)-(*E*)-hept-2-ene	25.1	15.9	78.8	3.1	[110]
	1-(4″-Methoxyphenyl)-7-(3′,4′-*di*-methoxyphenyl)-(*E*)-hept-2-ene	28.3	5.3	56.2	2.0	
*Acronychia crassipetala* (Rutaceae)	Acronyol A	31.4	45.3	108.0	3.4	[100]
	Acronyol B	55.4	82.6	114.0	2.1	
*Angophora costata* (Myrtaceae)	Costatamin A	89.5	91.1	>115.0	>1.3	[83]
	Costatamin B	67.7	115.0	>115.0	>1.7	
*Citrus garrawayi* (Rutaceae)	Garracoumarin C	44.0	15.4	<90	<2.1	[111]
	Garracoumarin D	53.9	<90	<90	<1.7	

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
