# Peer review of "From the Bush to the Brain: Preclinical Stages of Ethnobotanical Anti-Inflammatory and Neuroprotective Drug Discovery—An Australian Example"

_ijms, 2023, doi:10.3390/ijms241311086_

Round 1

Reviewer 1 Report

The review "From the bush to the mouse - preclinical stages of herbal drug discovery from an Australian perspective" by the group of Gerald Münch is a well writen paper which gives insights into the pathophysiology of Alzheimer especially the pathological changes of immun reactions and reviews recent findings about herabl drug discovery in Australia. From my point of view, the part dealing with methods and cell models to screen herbal compounds is of high interest for the scientific community.

Reviewer 2 Report

Overall, this is a good review manuscript, however, the following comments need to be addressed prior to publication:

1. Line 151-152, sentence need to be rewritten to be a complete statement.

2. Line 168, can author list a few details of what kind of SAIDs drugs were used in this study and how significant the association is with the lower risk of AD?

3. Line 172, NTFs duplications need to be fixed.

4. In 4.2, has the iPSC-derived microglial been used for the cell-based screening assays? If so, worth it including it to the section?

5. Organ on chip technology has played a part in drug screening.  Has any microfluidic chip model (e.g. organ-on-chip) been used for the initial drug screening?

6. Font size at Line 1024-1028 need to be fixed.

N/A

Reviewer 3 Report

The manuscript 'From the bush to the brain: preclinical stages of ethnobotanical drug discovery – an Australian example' reports a good level of data.

However, there are some comments regarding the text:

1.       Please, correct the title - for instance, like this: 'From the bush to the brain: preclinical stages of ethnobotanical data regarding neuroprotective drugs' discovery – an Australian example'

2.       The section "Introduction" should be added. The purpose of the work is not stated. It should show the relevance of research on plants of the Australian flora and their bioactive components, as stated in the title of the manuscript (because the word "plant" appears for the first time only on page 6 of the text of the article).

3.       I propose to introduce the main abbreviations Alzheimer's disease (AD), Parkinson's disease (PD), multiple sclerosis (MS) in the Abstract (not only for Alzheimer's disease).

4.       Figure 1 'Drug discovery pipeline for herbal compounds' is too general  and reflects the overall methodology of studying plants of folk medicine. In addition, a branch of common barberry is depicted at the beginning, which is not of Australian, but Eurasian origin.

5.       There is no reference to Figure 2 in the text.

6.       It is necessary to clearly indicate the scientific source from which it is clear that Curcuma longa  (as the source of key compound curcumin which was mentioned in the paper more than 40 times!) comes from Australia or the neighboring islands, but not from India  (https://en.wikipedia.org/wiki/Turmeric). And one more thing: why is there no data about this plant  in Table 1?

7.       Since apigenin is one of the key polyphenols whose properties are described in this article, it is worth adding information about its plant sources which originated from Australia: https://pubmed.ncbi.nlm.nih.gov/26304400/

8.       The Conclusions should be expanded to more fully reflect the entire scope of the analysis of literary sources.

9.       The main notes regarding use of the abbreviations:

·      Lines 59, 144, 152, 175, etc.it is no need to repeat the full name 'Alzheimer's disease' again and again as the abbreviation AD was used in the Abstract.

·      Lines 64, 430 - it is no need to repeat the full name central nervous system in the different places of the text as abbreviation CNS was used firstly in line 49.

·      Lines 75, 508 – the same point regarding abbreviation of 'advanced glycation endproducts (AGEs)'

·      Lines 131, 143 -  the same point regarding abbreviation of Non-steroidal anti-inflammatory drugs (NSAIDs),

·      generally, all abbreviations need to be checked for avoiding repeatings.

10.   The italic type should be used everywhere for the term 'In vivo' as well as for the Latin names of species for example: lines 252, 253, etc. - in vivo; line 1185 Melaleuca alternifolia, etc.

11.   There are more than 200 literary sources used for preparing this review manuscript. However, I propose to expand the discussion slightly, for example, regarding the properties of the main phytoconstituents as well as adding information about the Eucalyptus species that are endemic to the Australian flora. These can be found in the following and other sources:

·       Ali A, Bashmil YM, Cottrell JJ, Suleria HAR, Dunshea FR. LC-MS/MS-QTOF Screening and Identification of Phenolic Compounds from Australian Grown Herbs and Their Antioxidant Potential. Antioxidants (Basel). 2021 Nov 5;10(11):1770. doi: 10.3390/antiox10111770.

·       Anusha C, Sumathi T, Joseph LD. Protective role of apigenin on rotenone induced rat model of Parkinson's disease: Suppression of neuroinflammation and oxidative stress mediated apoptosis. Chem Biol Interact. 2017 May 1;269:67-79. doi: 10.1016/j.cbi.2017.03.016.

·       Quispe C, Cruz-Martins N, Manca ML, Manconi M, Sytar O, et al. Nano-Derived Therapeutic Formulations with Curcumin in Inflammation-Related Diseases. Oxid Med Cell Longev. 2021 Sep 15;2021:3149223. doi: 10.1155/2021/3149223. 

·       Vuong QV, Chalmers AC, Jyoti Bhuyan D, Bowyer MC, Scarlett CJ. Botanical, Phytochemical, and Anticancer Properties of the Eucalyptus Species. Chem Biodivers. 2015 Jun;12(6):907-24. doi: 10.1002/cbdv.201400327. PMID: 26080737.

Good level of English

Reviewer 4 Report

In the manuscript submitted for review, the authors have embarked on a scientific endeavor to provide a comprehensive and informative overview of the various research methods and techniques used in the analysis of plant materials from the rainforests. Their intention, in my opinion, was not only to present a wide range of available methodologies, but also to demonstrate their own research as an example of the successful application of these techniques. Focusing on the study of the complex realm of neuroinflammation, the authors delved into the complexity of this topic.

One of the strengths of the manuscript is the authors' skillful ability to explain the pharmacological basis of the analyses they presented. They meticulously reviewed the underlying mechanisms and principles, providing a solid foundation for understanding the research methods used. Their thoroughness in this regard is commendable, as it not only covers theoretical aspects, but also practical insights into the application of these methods.

The clarity and coherence of the manuscript is noteworthy, as the authors managed to present their findings in a clear and accessible manner. They skillfully articulated the multidirectional possibilities of using research on the activity of compounds and substances derived from plants within the context of traditional medicine.

While the manuscript as a whole is commendable, I have a few minor editorial issues that I would like to point out. In line 335, it would have been beneficial to italicize a phrase Hylandia dockrillii  to comply with established style guidelines. In addition, in Table 1, I suggest that botanical families be included or removed altogether.

According to me, the manuscript can make a valuable contribution to the analysis of  not only rainforest plants and have an impact on future scientific research and their practical applications.
